# Low-Toxicity Self-Photosensitized Biohybrid Systems for Enhanced Light-Driven H_2_ Production

**DOI:** 10.3390/ijms25063085

**Published:** 2024-03-07

**Authors:** Yuelei Wang, Yuqi Liu, Long Bai, Jueyu Wang, Na Zhao, Daizong Cui, Min Zhao

**Affiliations:** College of Life Science, Northeast Forestry University, Harbin 150040, China; wangyuelei1020@163.com (Y.W.); liuyuqi0921@126.com (Y.L.); bailongs@126.com (L.B.); 18804503512@163.com (J.W.); siyu19831114@163.com (D.C.)

**Keywords:** biohybrid systems, CdS nanoparticles, hydrogen production, cytotoxicity, peroxidase

## Abstract

Nanoparticles (NPs) represent a potential optoelectronic source capable of significantly boosting hydrogen production; however, their inevitable cytotoxicity may lead to oxidative damage of bacterial cell membranes. In this study, we employed non-photosynthetic *Escherichia coli* K-12 as a model organism and utilized self-assembled cadmium sulfide (CdS) nanoparticles to construct a low-toxicity and hydrogen-production-enhancing self-photosensitive hybrid system. To mitigate the cytotoxicity of CdS NPs and synthesize biocompatible CdS NPs on the cell surface, we employed engineered *E. coli* (*efeB*/OE) for bioremediation, achieving this goal through the overexpression of the peroxidase enzyme (EfeB). A comparative analysis with *E. coli*–CdS revealed a significant downregulation of genes encoding oxidative stress proteins in *efeB*/OE–CdS post-irradiation. Atomic force microscopy (AFM) confirmed the stability of bacterial cell membranes. Due to the enhanced stability of the cell membrane, the hydrogen yield of the *efeB*/OE–CdS system increased by 1.3 times compared to the control, accompanied by a 49.1% reduction in malondialdehyde (MDA) content. This study proposes an effective strategy to alleviate the toxicity of mixed biological nanoparticle systems and efficiently harness optoelectronic electrons, thereby achieving higher hydrogen production in bioremediation.

## 1. Introduction

In recent years, the increased consumption of fossil fuels has increased the emission of greenhouse gases and pollutants and has therefore had a huge impact on the environment and society [1,2]. To solve this problem, the development and utilization of renewable and clean energy sources, such as solar energy and hydrogen, have seen a renewed interest [3]. Notably, semiconductor nanomaterials can effectively harvest light energy and provide photoexcited electrons for power generation or hydrogen production [4]. A nonphotosynthetic bacterium, *Moorella thermoacetica*, precipitated with cadmium sulfide (CdS) quantum dots, is capable of selectively producing acetic acid upon irradiation [5]. Self-photosensitized hybrid systems are now widely used in several fields. In the hybrid systems, CdS nanoparticles (NPs) self-precipitated on the bacterial cell surface deliver electrons to intracellular pathways for production, thus enhancing biomass synthesis, energy production, and electrical energy generation [6]. Significant results have been achieved by using *Sporomusa ovata*–CdS for carbon fixation [7] and *Rhodopseudomonas palustris*–CdS for nitrogen fixation [8]. Inspired by this work, other CdS biohybrid systems have been developed for light-driven hydrogen production, such as *Desulfovibrio desulfuricans*–CdS [9], *Shewanella oneidensis*–CdS [10], and *E. coli*–TiO_2_ [11], providing new strategies for H_2_ production utilizing photoelectrons. Self-photosensitized hybrid systems, such as *E. coli*–CdS, generally improve H_2_ production by about 30% [12]. In such systems, the light-harvesting semiconductors, such as CdS [13] and dye-sensitized TiO_2_ [14,15] nanoparticles, endow the dark-fermentative bacteria with extra ability to utilize solar energy, thereby enhancing biological hydrogen production [11]. Importantly, many environmentally ubiquitous bacteria (e.g., *Clostridium butylicum* and *E. coli*) can be used to construct such inorganic-bio hybrids, and their potential application for wastewater treatment with enhanced denitrification [16], organic pollutant degradation [17,18], or heavy metal removal [19] has been demonstrated.

However, the efficiency of such biohybrid systems over long time periods remains to be investigated. Theoretically, NPs can be toxic to bacteria under visible light (VL) irradiation [20]. NPs containing heavy metal ions damage bacterial cells [21]. Bacteria are damaged by oxidative stress generated by NPs deposited on the surface or inside, such as CdS, Ag_2_S, and As_2_S_3_ [22,23,24]. Taking CdS NPs as an example, the toxicity can be divided into three types: Cd^2+^ generated by the dissociation of CdS NPs, CdS NPs, and oxidative holes generated by light radiation [25]. Biological functions are affected by the oxidative stress generated by NPs [26,27]. However, the introduction of photo-active NPs into the cells may also reduce the bacterial activity because of increased oxidative stress and photocatalytic production of reactive oxygen species [28,29,30]. Chen et al. constructed a two-hybrid system to reduce the cytotoxicity of CdS NPs to bacteria by synthesizing Mn_3_O_4_ NPs on the surface of the *Thiobacillus denitrificans*–CdS system [20]. Cui et al. reduced the cytotoxicity of the hybrid system by synthesizing CdSe_x_S_1−x_ inside *E. coli* cells, which inhibited ROS, thus enhancing hydrogen production [31]. These studies have focused on attenuating the cytotoxicity of nanoscale CdS particles toward bacterial cells through the modification or engineering of CdS. However, an alternative strategy that remains unexplored is the augmentation of intrinsic bacterial cell robustness as a potent means to enhance the photogenerated electron influx into bacteria. Therefore, the feasibility and optimal ways for constructing such a stable and efficient biotic–abiotic hybrid to produce H_2_ are still to be explored, and the underlying mechanisms of the nanoparticle–cell synergy to enhance fermentative hydrogen production should be clarified.

In this study, we constructed a stable hybrid photosynthetic system by coupling engineered *E. coli* with CdS NPs for hydrogen production. The impact of cytotoxicity induced by CdS nanoparticles in the biogenic hybrid system on bacterial cell membranes was investigated. The feasibility of the stable cell membrane for increased hydrogen production in the low-toxic engineered bio-CdS system was further verified. The photoelectric properties and cytotoxicity of the bio-NPs were explored. The effect of cell membrane changes on bacterial hydrogen production was analyzed and the mechanism underlying hydrogen production and metabolism in the hybrid system was determined.

## 2. Results

### 2.1. Construction and Characterization of the E. coli–CdS Hybrid System

To construct the *E. coli*–CdS system, cysteine and Cd^2+^ were co-incubated with stable-stage *E. coli* in the synthesis medium for 16 h. After the medium changed to bright yellow (Appendix A), TEM images of bacterial samples showed a gelatinous shell covering the cells, resulting in a rough surface (Figure 1a,b and Appendix A). Similarly, the CdS NPs deposited on the surface of *E. coli* were partly on the surface of the bacterium and partly embedded in the periplasm [12], with a particle size of approximately 15–25 nm (Appendix A). SEM results showed that most of the NPs were uniformly dispersed on the cell surface, while some parts were protruding and connected (Figure 1c,d and Appendix A). It was speculated that the aggregation of cysteine desulfurase led to the massive deposition of CdS NPs, although further studies are required for their definitive identification. Alternatively, the phenomenon can be explained by the formation of particle clusters by NP aggregation [16]. Moreover, the properties of EDS confirmed that the NPs on the surface of bacterial cells were Cd and S with an approximately 1:1 atomic ratio (Figure 1e).

Furthermore, three diffraction peaks of the XRD pattern were detected in the purified CdS NPs, showing three broad peaks with 2θ values of 26.546°, 44.039°, and 52.162° that could be indexed to the (111), (220), and (311) planes, respectively. The XRD pattern matched well with the CdS standard sample (JCPDS No. 80-0019) (Figure 1f), and no significant impurities were detected, indicating that the sample was highly pure. Further, the XRD spectral results revealed the cubic crystal structure of the biogenic CdS NPs, proving that the material immobilized on the surface of *E. coli* was CdS. Based on the UV–Vis absorption spectra, CdS NPs absorb light in the range of 450–550 nm (Figure 1g). The band gap of CdS NPs, calculated from the Tauc plot, was 2.54 eV (Figure 1h), indicating that the newly synthesized CdS NPs had good visible-light absorption ability. Taken together, these results confirm the successful biosynthesis of CdS NPs and support the feasibility of constructing a self-photosensitized photosynthetic system.

### 2.2. Toxicity Assessment of Cd^2+^ and CdS NPs on Whole Bacterial Cells

Under light radiation, Cd^2+^ was dissociated from CdS NPs on the bacterial cell surface in the presence or absence of cysteine (Figure 2a). Although the dissociation of CdS NPs was attenuated by cysteine as a hole reducer, the intracellular or extracellular Cd^2+^ content was significantly elevated (Figure 2b,c). It is worth noting that a similar reduction in the Cd^2+^ content was produced by CdS NPs with or without cysteine under dark conditions, demonstrating that the stability of CdS NPs is affected by light. Hence, the SOD, CAT, and POD enzyme activity levels within bacteria were examined in the presence of different toxic substances. SOD, CAT, and POD activity increased in bacteria due to the presence of Cd^2+^ (Appendix A), with greater increases in the activities of SOD and CAT than POD. The same results were obtained when the *E. coli* (dead)–CdS system was added to living *E. coli* (Appendix A). Oxidative holes generated by CdS NPs under light conditions trigger the *E. coli* self-defense mechanism. SOD, CAT, and POD enzyme activities increased, while POD showed minimal activity. These results suggested that the cytotoxicity of both Cd^2+^ and CdS NPs caused oxidative stress to *E. coli*.

To further assess the cytotoxicity of CdS NPs, oxidative-stress-related genes associated with *E. coli*–CdS were quantitatively evaluated. Various genes, such as superoxide dismutase (MN-*sodA*), catalase/hydroperoxidase HPI (*katG*), nucleic acid endonuclease IV (*nfo*), and NADP (+)-dependent glucose-6-phosphate dehydrogenase (G6PDH) (*zwf*), were selected for quantitative studies. The *sodA*, *katG*, *zwf*, and *nfo* genes were more highly expressed in *E. coli*–CdS under light conditions than in the control (Appendix A), indicating that the intracellular environment of *E. coli* was disturbed by the toxic effects produced by light exposure to CdS NPs. Thus, bacterial cells need to express a large number of antioxidant proteins to protect against oxidative damage [21]. These results indicated that the oxidative stress response and cellular stability of *E. coli* were affected by Cd^2+^ and photoexcited CdS NPs.

### 2.3. Construction of the Engineered efeB/OE–CdS System

The ability of bacterial cell membranes to transfer photoelectrons will be affected due to the large amount of toxic CdS NPs loaded on the cell surface. Accordingly, the peroxidase gene (*efeB*) was transferred into *E. coli* for stable intracellular overexpression to test its effect on the hybrid system with the goal of conferring protection against oxidative stress. A similar growth curve was observed for *E. coli* and *efeB*/OE, revealing that gene manipulation did not affect cell growth (Appendix A). Additionally, the concentration of Cd^2+^ in the medium was determined during CdS biosynthesis (Appendix A). After CdS biosynthesis, there was no detectable difference among strains, indicating that gene manipulation did not affect the bacterial CdS biosynthesis ability. According to TEM images (Figure 1b and Appendix A), no remarkable differences in terms of the localization and amount of CdS NPs were observed among *E. coli* and *efeB*/OE strains, and the loss of bacterial hairs and flagella on the surface of *efeB*/OE–CdS cells was observed. Meanwhile, in *E. coli*–Ag_2_S [32] and *Geobacter sulfurreducens*–CdS [17] systems, the bacteria exhibited the same phenomenon. Moreover, compared to *E. coli* (Appendix A), the bacteria introduced with the EfeB fusion protein showed a strong fluorescent signal under fluorescence microscopy, indicating that *efeB*/OE is an effective strain (Appendix A). These results suggested that EfeB overexpression did not affect the normal growth state of *E. coli*, and CdS NPs’ synthesis did not differ between the *efeB*/OE and normal strains. This laid the foundation for the stable production of hydrogen by electron transduction.

### 2.4. Contribution of EfeB to Light-Driven Hydrogen Production

To test the ability of EfeB in the *efeB*/OE–CdS system to enhance the resistance of bacteria to oxidative stress, the physiological properties of bacterial cells were examined under various conditions. The MDA content in *efeB*/OE–CdS cells was significantly lower (*p* < 0.05) than that in the control group as the duration of light exposure increased, indicating that the damage to *E. coli*–CdS cells was more severe (Appendix A). Oxidative damage to the bacterial cell membrane was reduced by EfeB, resulting in the concentration of MDA with a reduction of 49.1% in the *efeB*/OE–CdS system. Meanwhile, the cell membranes of biohybrid systems were identified by PI staining, which only penetrates cells with impaired membranes. After 12 h of light exposure, approximately 5% of the negative control *E. coli*–CdS (0 h) (Appendix A), 10% of *E. coli*–CdS (12 h) (Appendix A), and 0.8% of *efeB*/OE–CdS (12 h) (Appendix A), respectively, were stained with PI, indicating that the cell membrane was protected by EfeB, thereby contributing to the ability of the hybrid system to efficiently produce hydrogen.

To further verify the conjecture that the photoelectrons were not stably utilized by bacteria under illumination due to the damage caused by NPs to the cell membrane, cell membranes of *E. coli*–CdS and *efeB*/OE–CdS hybrids after light treatment were observed by AFM. The surface of normal *E. coli* is smooth and flat, and bacterial hairs and flagella are present (Appendix A). However, the surfaces of *E. coli*–CdS and *efeB*/OE–CdS hybrids without light treatment (0 h) already showed a small amount of damage (Appendix A), again indicating that the cell membrane was damaged by CdS NPs and Cd^2+^. After 12 h of photo-driven hydrogen production, the surface of the *E. coli*–CdS system became rough and was partially depressed in some areas due to severe damage (Figure 3a–c). In contrast, the *efeB*/OE–CdS cell membrane was nearly undamaged and the surface was very smooth and flat, thus providing a more stable state for the bacterial utilization of photoelectrons for hydrogen production (Figure 3d–f). Both the hairs and flagella on the surface of hybrid systems exposed to oxidative stress disappeared, consistent with the TEM and SEM results. These data confirmed that the membranes of our engineered *efeB*/OE–CdS hybrid under anaerobic conditions were extremely reliable for H_2_ production. Notably, the elevated expression of EfeB effectively reduced the phototoxicity of CdS NPs and created a favorable production environment for the hybrid system.

Immediately afterwards, oxidative-stress-related genes from the intracellular environment were measured under light conditions. During light radiation, compared with *E. coli*–CdS, the expression levels of *efeB* were considerably upregulated (Appendix A) and those of chaperone protein genes (*dnaK*) and *nfo* were significantly downregulated (*p* < 0.05) (Appendix A) in the *efeB*/OE–CdS hybrid. The expression levels of *sodA* and *katG* were also reduced (Appendix A); however, these reductions were not significant. These results indicated that EfeB effectively reduced the cytotoxicity of CdS NPs and maintained normal functions of the bacterial cells, including protein synthesis and gene expression. Interestingly, similar expression levels of cell division genes (*fstQ*) in *E. coli*–CdS and *efeB*/OE–CdS were observed (Appendix A). It was hypothesized that the toxicity of CdS NPs under light caused partial damage to cell membranes and did not affect bacterial growth or death. To clarify the speculation, bacterial growth in the biohybrid system under light was analyzed. Under VL irradiation (750 Wm^−2^), no significant difference between the growth status of *efeB*/OE–CdS and *E. coli*–CdS was observed (Appendix A), illustrating that bacterial survival was not affected by CdS NP toxicity.

To validate the hydrogen production ability of *efeB*/OE–CdS and *E. coli*–CdS, untreated *E. coli* and *E. coli* (dead)–CdS were selected as controls under VL irradiation. In the hybrid system, the photoelectrons generated by CdS NPs were absorbed by *E. coli*, resulting in increased H_2_ production. By using a glassy carbon electrode photoelectrochemical cell, the photoelectric properties of bio-NPs were examined. They exhibited significant photocurrent production during a light–dark cycle (Figure 4a), indicating robust photocatalytic activity for NPs, thereby providing light-harvesting ability. Finally, H_2_ production by *E. coli*–CdS and *efeB*/OE–CdS hybrids was significantly higher than that in the control after 12 h of light at a light intensity of 750 Wm^−2^ (Figure 4b). The hydrogen yields for untreated *E. coli* and *E. coli*–CdS were approximately 3.3 mmol and 11 mmol, respectively. The H_2_ generated by the *efeB*/OE–CdS system was approximately 13.4 mmol, resulting in an additional 2.4 mmol of H_2_ compared with that in the *E. coli*–CdS system (i.e., 4-fold the hydrogen production of *E. coli*). An AQE of 5.94% and 4.16% was achieved, for *efeB*/OE–CdS and *E. coli*–CdS systems, respectively, which is higher than that reported for most biohybrid systems with self-produced semiconductor nanoparticles (Appendix A). Moreover, the system demonstrated a remarkable stability. Evidently, the apparent quantum efficiency of the *efeB*/OE–CdS system surpasses that of the *E. coli*–CdS system by 1.78%. These results indicated that the photoelectrons released from CdS NPs under VL irradiation were efficiently utilized by the stable membrane system of *efeB*/OE–CdS hybrids, leading to increased hydrogen production.

To test the sustainable role of the *efeB*/OE–CdS system in promoting H_2_ production, a recycling experiment was carried out. After glucose supplementation at 12 h, the hydrogen production of *efeB*/OE–CdS and *E. coli*–CdS is shown in Appendix A. After 30 h of irradiation, the hydrogen yields of *efeB*/OE–CdS and *E. coli*–CdS were 25.3 mmol and 18.5 mmol, respectively, indicating that EfeB was capable of sustainably improving the use of photoelectrons by *E. coli*–CdS. This indicated that the *efeB*/OE–CdS demonstrated a remarkable stability for H_2_ production.

### 2.5. Mechanism Underlying Hydrogen Production in Light-Assisted Hybrid Systems

As shown in Figure 4c, no significant differences in glucose consumption were observed for *E. coli*–CdS and *efeB*/OE–CdS, indicating that EfeB overexpression did not affect the cellular utilization and uptake of glucose, ensuring that the glycolytic pathway proceeded normally. The *efeB*/OE–CdS system showed higher levels of both pyruvate and formate than those in the control (Figure 4d), suggesting that the solid cell membrane allowed a higher flow of photoelectrons into the pyruvate and formate synthesis pathways. The pyruvate content peaked at 3 h and then decreased until 12 h, indicating that the glycolytic pathway in the hybrid system reacts faster, owing to a lower toxicity in the early stage and, conversely, a slower reaction rate in the later stage (Figure 4e). A corresponding increase in lactate was observed in both the *E. coli*–CdS and *efeB*/OE–CdS systems after the lactate content in the lactate fermentation pathway was examined (Figure 4f). As the glycolytic reaction proceeded, formic acid was catabolized by FDH into NADH and carbon dioxide. The FDH activity and NADH/NAD ratio in the *efeB*/OE–CdS system were higher than those in the control (Figure 4g,h). These results showed that the stability of the *efeB*/OE–CdS system was better than that of *E. coli*–CdS, and FDH activity was further enhanced by the improved capture of photoelectrons, increasing the NADH/NAD ratio. Therefore, the increased level of pyruvate generated from an upstream pathway resulted in an increased formate concentration, further increasing hydrogen generation (Figure 4i). Overall, these results revealed that under VL irradiation, photoelectrons were efficiently exploited by the stable *efeB*/OE–CdS system, elevating the intracellular potential and thus inducing more hydrogen production from NADH, which was very beneficial for biological hydrogen production.

## 3. Discussion

In this work, CdS NPs were synthesized by *E. coli* in a medium containing cysteine and Cd^2+^ by exploiting the biosynthetic properties of *E. coli*. The successful synthesis of CdS nanoparticles on the cell surface was confirmed in TEM, SEM, EDS, XRD, and UV–Vis absorption spectroscopy characterization. The CdS NPs deposited on the surface of *E. coli* were approximately 15–25 nm in diameter, which is consistent with the majority of biosynthesized CdS NPs [9,12]. Interestingly, with the synthesis of CdS NPs, bacterial flagella and hairs disappeared, suggesting that the physiological properties of *E. coli* were affected by the deposition of CdS NPs. This phenomenon is consistent with the results observed in *M. thermoacetica*–CdS [5]. Toxic Cd^2+^ was converted into sulfide complexes by *E. coli* and deposited on the cell surface [33]. Given that Cd is a heavy metal element, the presence of Cd^2+^ ions can impact the normal physiological activities of bacteria. During the synthesis of CdS nanoparticles, an excessive concentration of Cd^2+^ ions can lead to bacterial death. However, an optimal concentration of Cd^2+^ ions can induce metabolic changes in bacteria, resulting in the transformation into less toxic CdS nanoparticles. Previous studies, such as the work by Wang et al., have optimized the Cd^2+^ ion concentration when synthesizing CdS nanoparticles using *E. coli*, achieving a balance that induces substantial synthesis of CdS nanoparticles without compromising bacterial viability [12]. Similarly, in hybrid systems like *T. denitrificans*–CdS [16], *D. desulfuricans*–CdS [9], and *Shewanella oneidensis*–CdS [34], these bacteria lose their flagella during the synthesis process. From this, it is inferred that the absence of flagella in bacteria serves as an indicator of Cd^2+^ ion toxicity leading to death, accompanied by distress and/or metabolic changes. Similarly, consistent with the results of several previous studies [5,9], TEM and SEM images (Figure 1b,d) revealed the precipitated CdS NPs on the cell surface, forming a relatively stable inorganic-bio system. 

In a normal manner, the mechanism underlying toxicity caused by *E. coli*–CdS was further evaluated. Biological processes can be affected by toxic NPs under light radiation [35]. Toxic Cd^2+^ is released by CdS NPs under illumination and enters bacterial cells through ion channels, resulting in physiological dysfunction [25]. This is confirmed by our findings in Figure 2. The photocorrosion of CdS NPs on the surface of *E. coli* under light radiation resulted in elevated intracellular and extracellular Cd^2+^ levels compared to the control under darkness. However, although the addition of cysteine attenuated the dissociation of CdS NPs, this phenomenon could not be completely avoided. It is speculated that this may be related to the crystal structure and light intensity of CdS NPs. In addition, Both CdS NPs and Cd^2+^ can affect the activity of bacterial cells [25]. Cytotoxicity from the oxidative dissociation of NPs exposed to air or light could increase in bacteria [36,37]. Thus, the levels of SOD, CAT, and POD enzyme activities in *E. coli* cells were significantly higher than those of the control due to the presence of Cd^2+^ and CdS NPs (Appendix A). Hossain et al. have demonstrated that the cytotoxicity of Cd^2+^ and CdS NPs affects the cellular activity of *E. coli* and HeLa cells, resulting in a significant increase in POD and SOD activities [35]. Due to photochemical degradation and the release of heavy metal ions, toxicity of nanomaterials has adverse effects on biological cells [38]. Furthermore, the expression levels of antioxidant genes as well as protein and DNA repair genes in bacterial cells were affected when *E. coli*–CdS was irradiated by light [21,39,40]. In *E. coli*, most genes induced by oxidative stress are grouped into two regulons: *SoxRS* and *OxyR*. Oxidized SoxR induces the expression of SoxS, which in turn activates the transcription of structural genes of the *SoxRS* regulon, including *sodA* and *zwf* [41]. Antioxidant enzymes, such as G6PDH, provide reducing power. Both Cd^2+^ and CdS NPs have been reported to affect the cellular antioxidant system by upregulating the expression of related genes, such as *ftsZ*, *mutS*, and *dnaK* [21]. Similarly, the expression levels of these genes (*sodA*, *katG*, *zwf*, and *nfo*) were higher in *E. coli*–CdS than in the control under light conditions (Appendix A), consistent with the cytotoxicity of ZnO [25]. The elevated expression levels of nfo indicated that bacterial gene transcriptional function was impaired (Appendix A). In conclusion, Cd^2+^, CdS NPs, and CdS NPs’ photooxidative toxicity affected the cellular activity of *E. coli*.

In *E. coli*–CdS hybrid systems, the cell membrane is susceptible to oxidative damage due to the large amount of CdS encapsulated on the bacterial surface, resulting in reduced photoelectron transfer efficiency. Protecting the stability of the cell membrane is key to enhancing the efficiency of hydrogen production. Previous studies have shown that bacterial cell membranes exposed to oxidative stress are effectively protected by EfeB [42]. Therefore, the *efeB* gene was transferred into *E. coli* species to enhance its expression level and protect the membrane system of *E. coli*–CdS. Interestingly, the ability of the *efeB*/OE strain to synthesize CdS was not affected. Consistently, the ability of the genetically modified *Shewanella oneidensis* MR-1 and *E. coli* BL21 to synthesize CdS was similarly unaffected [10,43]. Similarly, loss of cell surface hairs and flagella was observed in the *efeB*/OE–CdS system. In general, harmful substances have a detrimental effect on microorganisms, which damage the biological functions of cells [35,44].

MDA is produced by damaged cell membranes when bacteria are attacked by harmful substances, such as CdS NPs, Cd^2+^, and pollutants [45,46]. This means that lower levels of MDA indicate less damage, and a stable cell membrane will effectively ensure the normal operation of metabolic pathways [47]. Thus, in the *efeB*/OE–CdS system, we detected lower levels of MDA (Appendix A), indicating that the bacterial cell membrane was maintained. This could be that a higher amount of the peroxidase EfeB mediates the bacterial deep defense system, consistent with the results of *Streptococcus thermophilus* protecting human intestinal epithelial cells, HT-29 [42]. Similarly, the PI staining results confirmed the *efeB*/OE–CdS system with a stable cell membrane (Appendix A). Significantly, very few *efeB*/OE–CdS cells were observed to be stained red after light treatment (Appendix A), compared to the control. *E. coli* cells exposed to CdS conditions becoming more filamentous was reported, due to CdS interfering with the process of cell division [35]. Therefore, to observe the changes in bacterial cells more clearly, AFM experiments were performed (Figure 3 and Appendix A). The light-treated *E. coli*–CdS cell membranes exhibited large areas of damage compared to normal *E. coli*. In contrast, *efeB*/OE–CdS cell membranes were barely damaged and had a very smooth and flat surface. Also, both hairs and flagella on the surface of the hybrid system exposed to oxidative stress disappeared, which is consistent with the results of TEM (Figure 1). During light radiation, in the *efeB*/OE–CdS system, the *danK*, *nfo*, *sodA*, and *katG* all showed lower expression levels compared to the control (Appendix A). This may be due to the substantial upregulation of *efeB* expression levels stimulating the cells to readjust their defense system. Interestingly, there was little change in *fstQ* expression levels and bacterial cell growth numbers (Appendix A), indicating that CdS NP toxicity did not cause cell death. Likewise, previous work on H_2_ generated from hybrid systems did not report effects on cell death [12,31]. The deposition of CdS nanoparticles on the bacterial cell surface has resulted in the formation of a biohybrid system, wherein the functional processes do not induce bacterial lethality. As depicted in Appendix A, even after the reintroduction of glucose 12 h post-hydrogen production, the hybrid system maintains robust hydrogen generation activity, displaying a notable increase. Similarly, the *D. desulfuricans*–CdS system exhibits sustained hydrogen production over 240 h [9], underscoring the remarkable vitality and prolonged biological activity of the biohybrid system.

In the hybrid system, CdS NPs exhibited a strong photocurrent (Figure 4a), which is consistent with that generated in *T. denitrificans*–CdS [16]. This demonstrates that CdS can generate photoelectrons to promote hydrogen production through photoexcitation. The photoelectrons released by bio-NPs are utilized by most microorganisms to promote the production of metabolites or enhance degradation, such as hydrogen production, acetic acid synthesis, and azo dye degradation [5,9,17]. Hydrogen production was significantly enhanced in the *efeB*/OE–CdS system compared to the control, suggesting that the EfeB protected the cell membrane to produce a continuous and stable flow of photoelectrons. Moreover, the hydrogen yield of the *efeB*/OE–CdS system is higher than that reported for other biohybrid systems with self-produced semiconductor nanoparticles [9,10,12]. According to these analyses (Figure 4b), no hydrogen was generated from the heat-treated hybrid system, indicating that the surface CdS in the hybrid system could not produce hydrogen without the active biological system, and additional hydrogen was not directly generated from water electrolysis by the surface CdS [12]. This might be due, in part, to competitive binding by organic components of bacterial cells to hydrogen in water molecules, such as phospholipids in the membrane and peptidoglycan in the cell wall [48]. Interestingly, the toxicity of CdS NPs under light was also effectively reduced by Mn_3_O_4_ nanoenzymes, enhancing the stability of the biohybrid system [20]. However, further studies are needed to determine whether these two strategies can be combined to obtain better efficiency.

Under anaerobic conditions, glucose and CdS NPs were used as energy and electron donors for *E. coli* in a photo-driven hybrid system with the reaction pathway of hydrogen production by glycolysis. Glucose was catabolized and converted to pyruvate, which in turn produced formate [49]. To understand the pathway of photoelectron transfer, various metabolites were measured (Figure 4c–i). However, the increase in lactate was slower and smaller in the *efeB*/OE–CdS system, possibly due to the inhibitory effect of photoelectrons on lactate fermentation, commitment of pyruvate to hydrogen production, or lactate utilization as a hole scavenger [50]. To further reveal the mechanism, more detailed studies are needed in the future. *E. coli* possesses proton-reducing and formate-reducing pathways for hydrogen production [51], which both rely on NADH as an important reducing equivalent. Thereby, in the *efeB*/OE–CdS system, the high activity of FDH and the large amount of NADH stimulated more hydrogen production. Presumably, the protective effect of EfeB on the cell membrane is not only effective for *E. coli*–CdS hybrids but may also apply to other hybrid systems, such as *M. barkeri*–CdS for methane production and *R. palustris*–CdS for carbon sequestration [52,53].

Based on these results, a mechanism for the minimally toxic promotion of hydrogen production by the overexpression of EfeB in *efeB*/OE–CdS hybrids is proposed, as depicted in Figure 5. In the hybrid photosynthetic system, biogenic CdS NPs can absorb photons and generate photoexcited electrons/holes under VL illumination. Under oxidative stresses, such as Cd^2+^ and CdS NPs, oxidative cavities in the cell membrane of *E. coli* were damaged. This resulted in the upregulation of the intracellular expression of the oxidative-stress-related genes *sodA*, *katG*, *nfo*, and *danK*. However, a relatively stable production environment for biotic–CdS hybrid systems requires large amounts of EfeB, which facilitates electron transfer from abiotic materials. In parallel, the stable membrane system and intracellular reaction environment improved the flow of photoelectrons generated by CdS NPs into the glycolytic hydrogen production system. This increased the production of hydrogen. As a whole, the viability of cells in the *efeB*/OE–CdS system was enhanced without changing the original properties of CdS NPs, enabling a more efficient utilization of energy particles, reducing energy loss, and increasing hydrogen production. 

Through the introduction of engineered *E. coli* (*efeB*/OE) utilizing the peroxidase enzyme (EfeB), the system mitigates the cytotoxicity of CdS NPs and synthesizes biocompatible CdS NPs on cell surfaces. Compared to the *E. coli*–CdS system, the *efeB*/OE–CdS system exhibits a relative transcriptional downregulation post-irradiation, with atomic force microscopy (AFM) confirming membrane stability. Due to enhanced membrane stability, the *efeB*/OE–CdS system demonstrates a 1.3-fold increase in hydrogen production compared to controls, with a 49.1% decrease in MDA. Notably, the AQE of the *efeB*/OE–CdS system reaches 5.94%, surpassing AQE reported for most biohybrid systems with self-produced semiconductor nanoparticles [9,29,54]. This strategy, mitigating toxicity in bio-nanoparticle hybrid systems and effectively harnessing photoelectrons, presents a more feasible and economically efficient solution for the future of hydrogen production and bioremediation. We believe this research provides a sustainable technological pathway for future industrial applications, demonstrating outstanding performance in increasing hydrogen yields while addressing the inevitable cytotoxicity associated with nanoparticle exposure. We anticipate that our findings will offer valuable insights for further development and application in related fields.

## 4. Materials and Methods

### 4.1. Wild-Type and Genetically Engineered E. coli Strains

The wild-type *Escherichia coli* (strain K-12, substrain MG1655) strain was kindly provided by Prof. Yan Zhang (Tianjin University) [55]. To construct the strain with *efeB* (Gene ID: 946500) overexpression (*efeB*/OE), the target gene *efeB* was amplified, ligated, inserted into the pET28a plasmid (kana resistance), and introduced into *E. coli* with the receptor state. After two rounds of selection, the *efeB*/OE strain was obtained and validated by the polymerase chain reaction (PCR) using the forward and reverse primer pair. The primers used in this study are listed in Appendix A.

### 4.2. Construction and Characterization of Biohybrid Systems

*E. coli* K-12 and *efeB*/OE were grown in a Luria–Bertani medium at 37 °C with shaking at 130 rpm to the late stationary phase. The cells were collected by centrifugation (6500× *g*, 5 min) and washed three times with water. Then, the cells were transferred to synthetic medium MI (Appendix A) containing 0.4 mM Cd^2+^ and 1 mM cysteine under aerobic conditions. The color of the suspension changed from white to bright yellow after controlling the cell UV density (OD_600_) to approximately 1.2 and incubation at 37 °C with shaking for 16 h. The biohybrid system was centrifuged (4000× *g*, 5 min) and washed three times with deionized water. The characterizations of biohybrid systems were observed according to a previously described method [21]. The CdS NPs were dispersed with a mixture of 5% Nafion and anhydrous ethanol (1:1) and dropped uniformly on a circular glassy carbon electrode with a diameter of 3 mm. The platinum sheet electrode and the glycogen electrode were used as auxiliary and reference electrodes, respectively. The photocurrent (I-T) of CdS NPs under photoexcitation was detected with an electrochemical workstation (CHI-760E; Shanghai Chenhua, Shanghai, China).

### 4.3. Cd^2+^ and CdS Assay during Light Irradiation

*E. coli*–CdS hybrid suspensions were taken at different time points under VL irradiation for the determination of Cd atoms. The contents of Cd were determined according to a previously described method [21]. In brief, *E. coli*–CdS hybrid suspensions were taken at different time points under VL irradiation for the determination of Cd atoms. In a 50 mL hybrid biological system mixture, the total cadmium ion content in various samples was determined using an atomic absorption spectrometer (PinAcle 900T; PerkinElmer Co., Ltd., Waltham, MA, USA).

### 4.4. Cell Enzyme Activity and MDA Assay under Oxidative Stress

Superoxide dismutase (SOD) (G0101Fl; Geruisi-bio, Nanjing, China), catalase (CAT) (G0105F; Geruisi-bio), and peroxidase (POD) (G0107F; Geruisi-bio) kits were used to assess *E. coli* and *E. coli*–CdS enzyme activity during exposure to oxidative stress. Under VL radiation, *E. coli*–CdS and *efeB*/OE–CdS hybrids produced by cell membrane damage after different light treatment durations were evaluated using the malondialdehyde (MDA) (G0109W, Geruisi-bio) kit, utilizing *E. coli*–CdS as a control group. All of the experiments were conducted following the procedures provided in the technical bulletins of the respective assay kits.

### 4.5. Quantitative Genetic Testing

Samples of *E. coli*–CdS were taken at regular intervals under light conditions and stored in a refrigerator at −80 °C. Total RNA was obtained using a Bacterial RNA Kit (R6950; OMEGA, Biel, Switzerland), following the manufacturer’s instructions. The concentration of the final extracted RNA was measured using the NanoDrop 2000 (DS-11; DENOVIX, Wilmington, DE, USA), and the RNA concentration of each sample was adjusted to 1 μg. Then, the total RNA was reverse transcribed into cDNA using the PrimScript™ RT Kit (Takara, Dalian, China) via the random primer method. cDNA obtained using the LightCycler 480 system with a Green Fast qPCR Mix (RR820A; Takara) was used for RT-PCR. For each sample, the average of three measurements was used to calculate relative abundance. The *16S rDNA* in each sample was selected as an internal reference. The relative abundance of each gene in the control was set to 1.0. The RNA primers used in this study are listed in Appendix A.

### 4.6. Cell PI Staining and Atomic Force Microscopy Analyses of Cell Surfaces

The extent of damage to the bacterial cell membrane was observed by propidium iodide (PI) staining experiments. The cells were centrifuged (4 °C, 1000× *g*, 2 min) and washed three times with 0.01 mM PBS. Then, 500 μL of the prepared PI staining working solution was added to each cell sample, and the cell precipitate was gently resuspended and placed in a water bath at 37 °C for 30 min protected from light. Images of PI-stained cells were obtained using a fluorescence microscope (BX41; OLYMPUS, Tokyo, Japan). The bacterial cell membranes exposed under oxidative stress conditions were examined utilizing an Atomic Force Microscope (AFM). Initially, normal *E. coli* cells, as well as *E. coli*–CdS and *efeB*/OE–CdS cells treated with 12 h of illumination, were collected and fixed overnight at 4 °C with 0.25% glutaraldehyde. Subsequently, the glutaraldehyde fixative was removed by centrifugation, and the specimens were rinsed three times with deionized water. A 10 µL suspension of bacterial cells was then dropped onto a silicon wafer, followed by air-drying at room temperature. AFM was employed to capture the morphological features and membrane structures of the bacteria. All AFM measurements were conducted in contact mode at room temperature in ambient air.

### 4.7. Light-Assisted Fermentative Hydrogen Generation

To evaluate the protective effect of peroxidase (EfeB) enzyme overexpression on the cell membrane on hydrogen production in *E. coli*, irradiation under a xenon lamp was performed. For the heat-treated group, the hybrid system was heat-treated in a 100 °C water bath for 20 min. To the 600 mL reaction vessels, 300 mL of hydrogen production medium MII (Appendix A) was added with 1 mM cysteine, while 0.5 mM IPTG inducer was added to the *efeB*/OE–CdS system. Then, the reaction vessel was anaerobically capped with high-purity N_2_ gas. Unless otherwise stated, the external light source was a 300 W xenon lamp (CEL-LAX; Ceaulight, Beijing, China), and the experiments were carried out in a water bath shaker at 37 °C with 12 h of light (750 Wm^−2^). The hydrogen content in the reaction vessel was measured periodically using a gas chromatograph (Model GC7890A; Agilent, Santa Clara, CA, USA).

### 4.8. Metabolite Assay in Biohybrid Systems during Hydrogen Production

Under natural light simulated using a xenon lamp, many metabolites were involved in the hydrogen production process. Therefore, we quantified these metabolites separately. Lactate and formate were measured by high-performance liquid chromatography (HPLC-1260 Infinity II; Agilent Technologies, Böblingen, Germany). Pyruvate (PA; G0870F, Geruisi-bio) and formate dehydrogenase (FDH) (G0445W; Geruisi-bio) quantification kits and NADH/NAD (G0801F, Geruisi-bio) kits were used to determine the concentrations of relevant intracellular compounds. All experiments were conducted following the procedures provided in the technical bulletins of the respective assay kits. All the above metabolites were produced by the catabolism of glucose, and the amount of glucose remaining at different time points was also monitored by the 3,5-dinitrosalicylic acid colorimetric method [56]. Finally, absorbance was recorded with a UV–Vis spectrophotometer and the glucose concentration was calculated according to a standard curve for glucose solutions.

### 4.9. Analytical Methods

The apparent quantum efficiency (AQE) was determined according to a previously described method [9]. The AQEs of the hybrid systems were determined, under LED illumination (λ = 420 nm and an irradiance of 20 mW cm^−2^). The irradiation area was 9.2 cm^2^. In these conditions, the *E. coli*–CdS and *efeB*/OE–CdS can produce 144.866 and 207.068 µmol of hydrogen after 3 h, respectively.

## 5. Conclusions

A stable *efeB*/OE–CdS system with low toxicity was constructed, combining genetic technology and the bacterial synthesis of NPs to enhance biological hydrogen production. This bio-inorganic hybrid system improved the photosynthetic efficiency of *E. coli* for the conversion of solar to chemical energy. Under VL irradiation, the cytotoxicity of Cd^2+^ and CdS NPs and oxidized cavities produced by CdS NPs were significantly reduced by an excess of EfeB, leading to a reduction in MDA. The photoelectrons generated by CdS NPs could better flow into cells with a stable membrane system in *efeB*/OE–CdS hybrids, providing a solid and stable foundation for an intracellular hydrogen production system. After 12 h of light exposure, H_2_ production in the *efeB*/OE–CdS system was 13.4 mmol, which was about 4-fold higher than that of *E. coli* and 2.4 mmol more than that of the *E. coli*–CdS system. In the hydrogen production process, the lower consumption of energy material was mainly attributed to the photoelectrons provided by the bio–CdS hybrid. This biotic−abiotic hybrid system based on bacterial surface display could be integrated with the natural environment to produce economically efficient clean energy, such as hydrogen, from wastewater and solar energy. To make a hybrid system suitable for many harsh environments, more efforts should be made in the future to continuously explore the protection of membrane systems by bioenzymes and its contribution to H_2_ production.

## Figures and Tables

**Figure 1 ijms-25-03085-f001:**
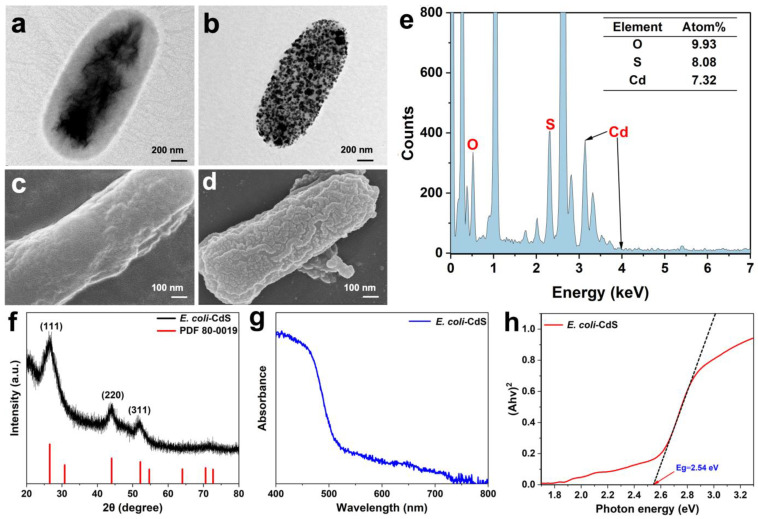
Electron microscopy of *E. coli* and *E. coli*–CdS hybrids. TEM images of *E. coli* (**a**) and *E. coli*–CdS hybrids (**b**). SEM image of *E. coli* (**c**) and *E. coli*–CdS hybrids (**d**). (**e**) The result of EDS analysis of the precipitated CdS nanoparticles. (**f**) XRD pattern of *E. coli*–CdS (reference peak from JCPDS data card No. 80-0019). (**g**,**h**) UV–Vis absorption spectra and Tauc plots of the hybrid systems.

**Figure 2 ijms-25-03085-f002:**
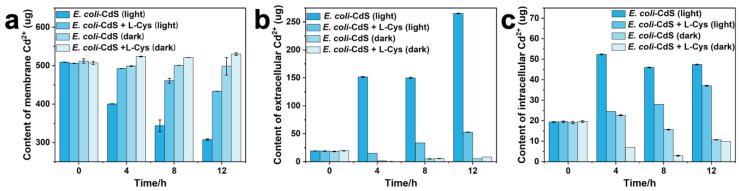
Leakage of cadmium ions from *E. coli*–CdS hybrids. After the photocatalytic reaction time, being 12 h, the content of (**a**) cell membrane Cd^2+^, (**b**) extracellular Cd^2+^, and (**c**) intracellular Cd^2+^.

**Figure 3 ijms-25-03085-f003:**
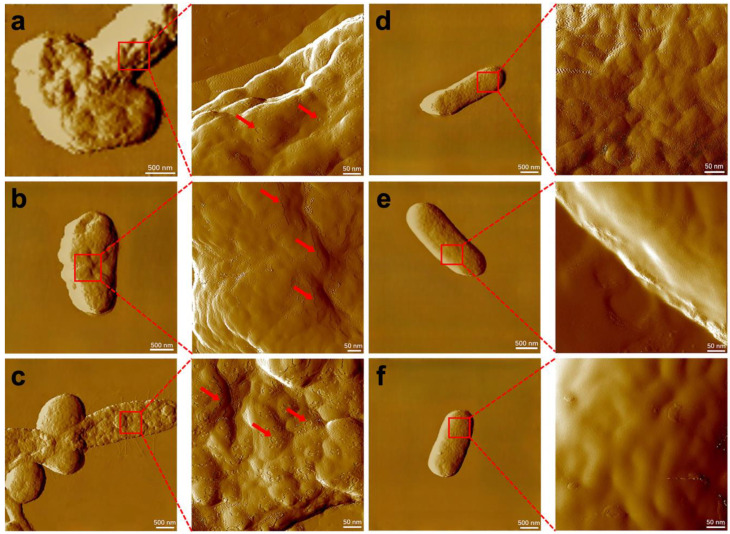
AFM study of the cell surface. AFM study of the surface topology of (**a**–**c**) *E. coli*–CdS and (**d**–**f**) *efeB*/OE–CdS cells after 12 h light illumination. Magnified portions are ultrastructural images. Red arrows, indicating areas of bacterial cell damage.

**Figure 4 ijms-25-03085-f004:**
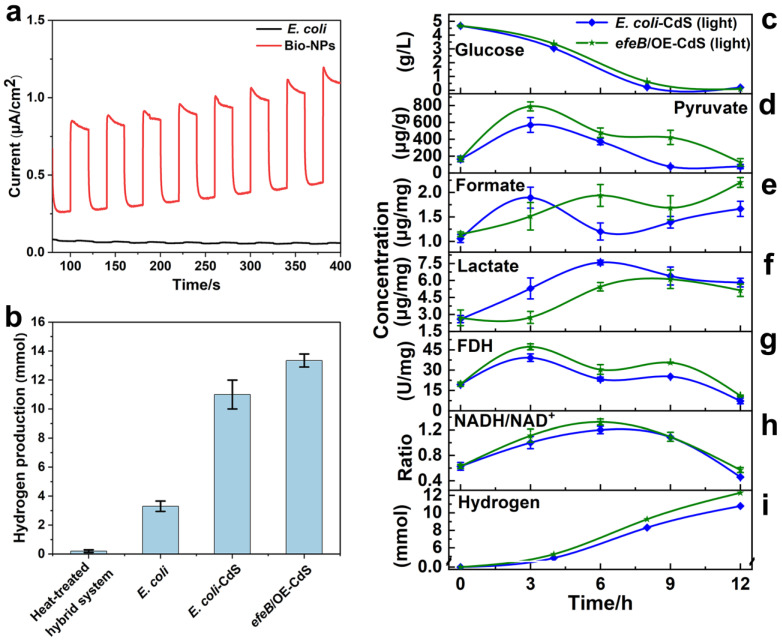
Light-driven hydrogen production ability of bacteria. (**a**) Instantaneous photocurrent of *E. coli* and bio-CdS NPS (extracted) under 750 Wm^−2^ visible light illumination during light on/off cycles. (**b**) Normalized hydrogen production in different test groups under 12 h irradiation. (**c**–**i**) Glucose and metabolite concentrations.

**Figure 5 ijms-25-03085-f005:**
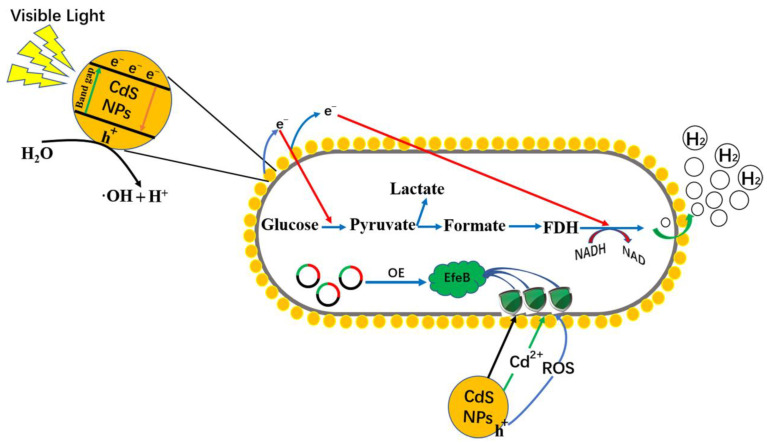
Proposed mechanism for enhanced photo-assisted hydrogen production by *efeB*/OE–CdS hybrids.

## Data Availability

The data presented in this study are available on request from the corresponding author.

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
