# Peer review of "Low-Toxicity Self-Photosensitized Biohybrid Systems for Enhanced Light-Driven H2 Production"

_ijms, 2024, doi:10.3390/ijms25063085_

Round 1
Reviewer 1 Report
Comments and Suggestions for Authors|
MS No:
|
ijms-2848184-peer-review-v1 |
|
Title:
|
Low-Toxicity Self-Photosensitized Biohybrid Systems for Enhanced Light-Driven H2 Production
|
|
Authors:
|
Yuelei Wang, Yuqi Liu, Long Bai, Jueyu Wang, Na Zhao, Daizong Cui, and Min Zhao
|
The present manuscript studies the efficiency of a self-photosensitized hybrid system for hydrogen production, by self-assembling cadmium sulfide (CdS) NPs in non-photosynthetic Escherichia coli K-12 as a model. I recommend it for publication on International Journal of Molecular Sciences after major revision.
Below are some specific comments:
· The Introduction needs revision in order to include more relevant citations. Moreover, the novelty of the present study should be pointed out.
· Abstract needs revision in order to be more specific.
· Cadmium leaching in the liquid phase should be quantified.
· A Table comparing the efficiency of the present system with other similar, should be added to the revised version.
· What about the practical value of the present study. A relevant discussion should be added in the revised manuscript.
Reviewer 2 Report
Comments and Suggestions for Authors
The manuscript deals with the preparation of E. coli bacteria coated by CdS nanoparticles to be used for the photochemical synthesis of hydrogen. CdS structural/morphological information, citotoxicity and cell damage information, hydrogen production activity of the system in 12h, improved membrane stability images, aerobic/anaerobic system behaviour, etc. are given. Finely, this study provides a strategy to mitigate the toxicity of bio-nanoparticles hybrid system and suggests an efficient use of their photoelectrons.
This manuscript collects a number of relevant technical results on the treated topic, however this information should be completed by further data before manuscript acceptance for publication. In particular, the following details about the synthesized CdS nanoparticles should be added:
- The average size of the CdS nanocrystals represents a relevant information and it should be provided. Owing to the CdS nanoparticle sintering (coalescence) on the bacteria surface, such information cannot be obtained by the microscopical investigation performed by SEM, but it can be calculated from the XRD by using the Scherrer equation. Please, calculate such value.
- It can be useful to provide some photos of the modified E. coli cultures (E.coli/CdS system) and also an optical microscopy (OM) image of them, for example obtained by the fluorescence microscopy technique.
- Since CdS nanoparticles (quantum-dots) have strong fluorescence characteristics, a fluorescence spectra (excitation and emission spectra) of the produced E. coli/CdS nanoparticles system should be provided in addition to the optical absorption spectrum.
- Is the absence of flagella in the bacteria an index of pre-death suffering and/or metabolic alteration due to the Cd2+ ion toxicity? Such point seems important; please, provide some comments about in the manuscript.
- Experimental details on the AFM sample preparation are missed; these details must be provided in the revised manuscript version.
- Concerning the CdS-coated bacteria lifetime, further information/discussion about should be provided in the revised manuscript version.
Comments on the Quality of English Language
English is quite fine, only minor editing is required.
Reviewer 3 Report
Comments and Suggestions for Authors
In this article, the Authors constructed a self-photosensitized hybrid system with low toxicity and enhanced hydrogen production activity by self-assembling cadmium sulfide nanoparticles in nonphotosynthetic Escherichia coli K-12 as a model. The research design is appropriate and well described. The results are clearly reported and support the conclusions. In any case, in my opinion, the real advantages of this device are not very clear. The Authors should better explain possible future applications also in terms of cost/benefit ratio.
Round 2
Reviewer 1 Report
Comments and Suggestions for Authors
Accept in present form